# May the Forgetting Be with You: Alternate Replay for Learning with Noisy Labels

## Abstract

Forgetting presents a significant challenge during incremental training, making it particularly demanding for contemporary AI systems to assimilate new knowledge in streaming data environments. To address this issue, most approaches in Continual Learning (CL) rely on the replay of a restricted buffer of past data. However, the presence of noise in real-world scenarios, where human annotation is constrained by time limitations, frequently renders these strategies vulnerable. In this study, we address the problem of CL under Noisy labels (CLN) by introducing **Alternate Experience Replay** (**AER**), a novel strategy that *takes advantage of forgetting* to maintain a clear differentiation between clean, complex, and noisy samples in the memory buffer. The idea is that complex or mislabeled examples, which hardly fit the previously learned data distribution, are the ones most likely to be forgotten. To grasp the benefits of such a separation, we equip AER with **Asymmetric Balanced Sampling**: a new sample selection strategy that prioritizes purity on the current task while retaining relevant samples from the past. Through extensive computational comparisons, we demonstrate the effectiveness of our approach in terms of **both accuracy and purity** of the obtained buffer, resulting in a remarkable average gain of 7.45% points in accuracy w.r.t. existing loss-based purification strategies.

## 1 Introduction

Despite the latest breakthroughs, modern AI still struggles to learn in a continuous fashion and suffers from *catastrophic forgetting* (McCloskey & Cohen (1989)), *i.e.* the latest knowledge quickly replaces all past progress. Therefore, Continual Learning (CL) has recently gathered an increasing amount of attention: among the others, one prominent strategy is to interleave examples from the current and old tasks (*rehearsal*). To do so, a small selection of past data is retained in a memory buffer (van de Ven et al. (2022); Chaudhry et al. (2019)), as in Experience Replay (ER) (Ratcliff (1990); Robins (1995)).

Intuitively, the effectiveness of these methods depends strictly on the content of the memory: the *larger* the gap between the memory and the true distribution underlying all the previous tasks, the *lower* the chances of learning a reliable model. In this respect, several factors may intervene and degrade the snapshot portrayed by the buffer. Several works have highlighted the shortcomings of low-capacity buffers and their link to severe overfitting (Verwimp et al. (2021); Bonicelli et al. (2022)). More recently, the plausible presence of annotation errors has emerged as an engaging factor (Kim et al. (2021); Bang et al. (2022)), due to the subsequent poisoning the memory buffer would be subject to. Indeed, not only a few observations of past tasks would be available for the learner, but they might be even erroneously annotated. It is noted that the presence of *noisy annotations* (Xiao et al. (2015); Lee et al. (2018); Li et al. (2017)) is an inescapably characteristic of CL: to allow the learner to digest incoming training examples on-the-fly, data has to be annotated within a restricted temporal window, leading to both poor human annotations and hasty quality controls.

In light of this, preliminary works (Kim et al. (2021); Bang et al. (2022)) focus on purifying the memory buffer, which allows these approaches to favourably consolidate their knowledge at the end of the current task (or while learning the task itself (Karim et al. (2022))). To do so, they spot clean samples by leveraging the popular *memorization effect* (Arpit et al. (2017); Han et al. (2018); Jiang et al. (2018)), stating that the most trustworthy examples are those favoured during the first

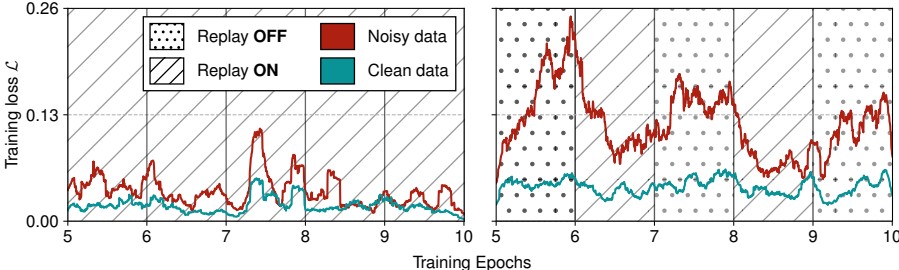

Figure 1: A visualization of the *training loss* of clean and noisy *buffer samples* during the *second task* of Seq. CIFAR-10 with $40\%$ noise. Standard replay makes the two lines indistinguishable (*left*) while alternating epochs of replay and forgetting result in a significant and persistent loss separation (*right*).

training stages, and hence they tend to exhibit a lower value of the loss function. However, despite its effectiveness in the offline scenario, such a criterion may be weak in CL: as learning does not re-start from scratch but builds upon previous knowledge, the adaption is faster and hence the loss-value separation between clean and noisy samples tends to vanish (Zhang et al. (2017); Arpit et al. (2017)).

To overcome this limitation, our work explores a radically different approach, which could be summarized by a quote ascribed to Julius Caesar "*si non potes inimicum tuum vincere, habeas eum amicum*". While existing methods see *forgetting* only as an issue to solve, we instead use it to identify noisy examples within the data stream. Indeed, we build upon the work of Toneva et al. (2019); Maini et al. (2022), which shows that noisy samples are highly prone to be forgotten. In particular, the notable work of Maini et al. (2022) mathematically proves that mislabeled examples exhibit rapid forgetting, whereas complex or rare instances are retained for longer periods (or not forgotten at all).

To illustrate such a phenomenon, we depict the loss trend for both the clean and noisy samples in a memory buffer produced by a rehearsal baseline (ER-ACE (Caccia et al. (2022))). In particular, Fig. 1 (left) shows the loss value sampled during standard training; differently, in Fig. 1 (right) we alternatively switch replay regularization on and off at each epoch. As can be seen, stopping replay has a distinct impact: while the loss value of clean samples remains low, it hugely increases for mislabeled examples. We remark that such a gap – in line with the results of Sec. 4.3 of Maini et al. (2022) – is maintained even when replay regularization turns on, as the model easily adapts to clean samples and hence learns them faster (Arpit et al. (2017); Jiang et al. (2018); Wei et al. (2020)).

In light of this, our main contribution is the introduction of **Alternate Experience Replay** (**AER**), a novel CL optimization scheme that alternates steps of **buffer learning** and **buffer forgetting** to encourage the separation of clean and noisy samples in the buffer. To the best of our knowledge, our work is the first one that exploits *forgetting* to purify the memory buffer while learning from an online stream. Furthermore, to take advantage of the enhanced separation brought by AER, we propose **Asymmetric Balanced Sampling** (**ABS**): a new sample selection strategy designed to select only clean samples while keeping the most informative samples from the past. By means of extensive experiments, we show that our proposals vastly improve the model's stability and performance, and can be easily applied to other rehearsal-based methods.

## 2 RELATED WORKS

### 2.1 CONTINUAL LEARNING

Continual Learning methods can be broadly categorized into *regularization-based* – these combat forgetting by limiting changes to key task-related parameters (Kirkpatrick et al. (2017); Zenke et al. (2017)) – and *rehearsal-based* methods (van de Ven et al. (2022)).

**Rehearsal** Complementing the current training data with samples from the past has been shown to prevent the forgetting of previous tasks better than any of the regularization-based methods in most existing CL scenarios (Chaudhry et al. (2019); van de Ven et al. (2022); Buzzega et al. (2020a)). A simple yet effective method is Experience Replay (ER) (Ratcliff (1990); Robins (1995)) which consists in interleaving the current training batch with past examples. Otherwise, Greedy Sampler

and Dumb Learner (GDumb) (Prabhu et al. (2020)) takes this idea to the extreme by greedily storing samples as they come and then trains a model from scratch using only samples inside the buffer.

**Sampling strategies** Given their low capacity, buffers need to contain a balanced outlook of all seen classes. For this purpose, many employ *reservoir sampling* (Vitter (1985)) to update the memory (Buzzega et al. (2020a); Bang et al. (2021); Caccia et al. (2022)). The outcome is an i.i.d. snapshot of the incoming tasks. However, not every sample comes with the same significance or robustness against forgetting, as highlighted by Buzzega et al. (2020b); Bang et al. (2021); Aljundi et al. (2019). Indeed, in Buzzega et al. (2020b); Bang et al. (2021); Aljundi et al. (2019) the authors show that retaining complex samples is crucial for preserving the performance, which they detect through their loss value or model uncertainty, respectively.

## 2.2 Learning with Noisy Labels

A popular approach for identifying noisy data is grounded on the memorization effect (Arpit et al. (2017); Jiang et al. (2018)), according to which correctly labeled (*clean*) instances tend to produce a smaller loss than mislabeled (*noisy*) ones during the initial stages of training. However, as training ensues and the model starts to learn wrong patterns from noise, its predictions become more unreliable (*confirmation bias*). In this regard, Han et al. (2020) rely on an explicit gradient ascent objective on the noisy samples, building on top of existing *sample selection* strategies or enhancing *loss correction* algorithms. Other works exploit two separate networks to address the sample selection phase and train only on a clean subset (CoTeaching (Han et al. (2018)), MentorNet (Jiang et al. (2018))) or on all seen samples with semi-supervised objectives (DivideMix (Li et al. (2020)), Arazo et al. (2019)).

## 2.3 Continual Learning under Noisy Labels

Recent studies (Bang et al. (2022); Kim et al. (2021); Karim et al. (2022)) conducted in the online CL setup have shown that existing sampling strategies fail to produce meaningful gains in noisy scenarios. In this respect, PuriDivER (Bang et al. (2022)) proposes a sampling strategy that promotes a trade-off between *purity* and *diversity* for samples in the buffer. Methods like SPR (Kim et al. (2021)) and CNLL (Karim et al. (2022)) use multiple buffers to gradually isolate clean samples: a *delayed* buffer temporarily collects data from the stream, and only clean examples are transferred to a *replay* or *purified* buffer. SPR trains a network using a self-supervised loss on samples from both buffers, while CNLL adopts a semi-supervised approach inspired by FixMatch (Sohn et al. (2020)).

## 3 Method

### 3.1 Problem setting

We formalize the problem of **Continual Learning** as learning from a sequence of $T$ tasks. During each task $t \in \{0, 1, \ldots, T\}$, input samples $\mathbf{X_t}$ and their annotations $\mathbf{Y_t}$ are drawn from an i.i.d. distribution $\mathcal{D}_t$. We follow the well-established class-incremental scenario (van de Ven et al. (2022); Farquhar & Gal (2018); Buzzega et al. (2020a)) in which $\tilde{\mathbf{Y}}_i \cap \tilde{\mathbf{Y}}_j = \emptyset$ and at task $t$ the learner $f_\theta$ is required to distinguish between all observed classes. Ideally, we wish to minimize:

$$\theta^* = \operatorname*{argmin}_{\theta} \mathbb{E}_t \left[ \mathbb{E}_{\mathcal{B} \sim \mathcal{D}_t} \left[ \mathcal{L}(f_\theta(\mathbf{x}), y) \right] \right], \tag{1}$$

where $\mathcal{L}$ is the cross-entropy loss and $\mathcal{B} = (\mathbf{x}, y)$. As in CL the objective above is inaccessible, we leverage a fixed-size buffer $\mathcal{M}$ to store and replay part of the incoming samples.

Formally, the generalized learning objective for rehearsal CL can be defined as:

$$\theta^* = \operatorname*{argmin}_{\theta} \mathbb{E}_{(\mathbf{x}, y) \sim \mathcal{D}_t} \left[ \mathcal{L}(f_\theta(\mathbf{x}), y) \right] + \mathcal{L}_R, \tag{2}$$

where the *replay regularization* term $\mathcal{L}_R$ depends on the choice of the replay-based method.

In this work, we start from the simple form of Experience Replay(Ratcliff (1990); Robins (1995)):

$$\mathcal{L}_R = \mathbb{E}_{(\mathbf{x}_r, y_r) \sim \mathcal{M}} \left[ \mathcal{L}(f_\theta(\mathbf{x}_r), y_r) \right]. \tag{3}$$

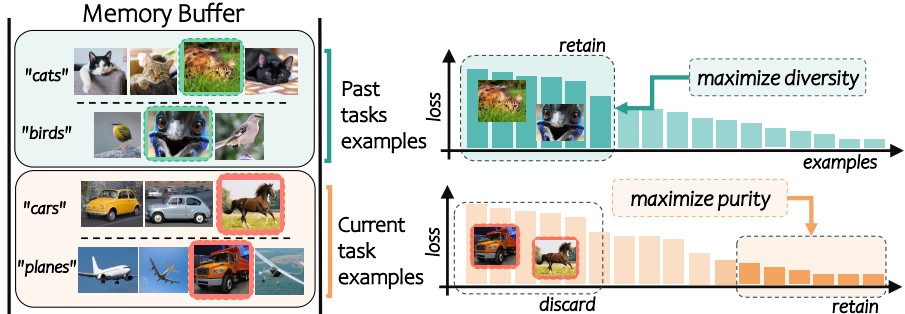

Figure 2: Depiction of Asymmetric Balanced Sampling (ABS). Samples of past classes are chosen to retain the most complex ones (*i.e.* maximize diversity to better counter forgetting), while we reverse the objective for the current task to maximize purity.

However, we wish to note that our proposal does not depend on the underlying choice of rehearsal CL and can be easily applied to more sophisticated choices of $\mathcal{L}_R$, as shown in Sec. 5.3.

Precisely, in this manuscript, we focus on the **offline** continual setting and tackle the challenge of **Continual Learning under Noisy Labels** (**CLN**), wherein the dataset is contaminated with spurious annotations to simulate noise in the data labelling process. For a given instance $\mathbf{x}_i \in \mathcal{D}_t$, we indicate with $\tilde{y}_i$ the label corrupted with annotation noise and with $\Pr(\tilde{y}_i \neq y_i)$ the respective *noise rate*.

## 3.2 ALTERNATE EXPERIENCE REPLAY

Our desiderata is the construction of a memory set $\mathcal{M}$ as clean as possible to address the performance degradation caused by $\tilde{\mathbf{Y}}_t$ and *mitigate forgetting*. For this purpose, we wish to maintain a strong gap between the loss of clean and noisy samples to facilitate sample selection based on the popular *small-loss criterion*. Indeed, with no countermeasure, the loss gap starts to deteriorate as the replay of a small selection of data ensues (Fig. 1a). This effect is even more pronounced in the popular *offline* (*i.e.* multi-epoch) CL setting (Rebuffi et al. (2017); Wu et al. (2019); Buzzega et al. (2020a)), where we might be forced trade-off convergence on the current task to avoid overfitting the mislabeled samples (Zhang et al. (2017); Arpit et al. (2017)).

To mitigate such an effect, we propose a new learning methodology named **Alternate Experience Replay** (**AER**), which encourages the separation between the losses of noise and clean samples by simulating *forgetting* of the buffer itself. Specifically, we distinguish between **buffer learning** and **buffer forgetting** epochs. During the former, we train the model with Eq. 2 while keeping the samples in $\mathcal{M}$ fixed. Then, during the latter, we disable $\mathcal{L}_R$ and train only on data from $\mathcal{D}_t$ to *encourage* forgetting. Exclusively in this specific stage, we enable sample selection to update $\mathcal{M}$ and take advantage of the different learning dynamics between clean and noisy samples (Fig. 1b). With the help of the sample selection strategy presented in Section 3.3, after the buffer forgetting epoch, we obtain a cleaner version of the memory buffer. However, simply cycling between buffer learning and forgetting results in the buffer being under-optimized, as it is effectively exploited during the former epochs. We avoid this by restoring $f_\theta$ to its previous state at the end of each forgetting epoch.

Finally, as noted by Ahn et al. (2021), the objective in Eq. 3 promotes the accumulation of bias towards the current task, which might skew the estimation of the loss and hamper sample selection. To mitigate this, we adopt the *asymmetric cross-entropy* of Caccia et al. (2022) to separate contributions of data from the incoming stream from those in $\mathcal{M}$.

## 3.3 SAMPLE SELECTION

### 3.3.1 SAMPLE INSERTION

At this stage, we prioritize mitigating the impact of noise on model training, leaving the task of determining the elements to retain to the sample replacement strategy introduced thereafter. In particular, when choosing which samples to include in the buffer, we perform an initial sample selection on the current batch of data $\mathcal{B}$. Let $\alpha$ be a threshold, we compute $\mathcal{R} = \{(\mathbf{x}, \tilde{y}) \in \mathcal{B} : \mathcal{L}(\mathbf{x}, \tilde{y}) < \alpha\}$. In our experiments, we set $\alpha$ as the 75th percentile of the loss computed on $\mathcal{B}$.

### 3.3.2 SAMPLE REPLACEMENT

**Asymmetric sampling**. Ideally, an effective strategy for sample selection should preserve the most representative samples from the past, while at the same time favour the release of mislabeled samples from the current task. As observed in Bang et al. (2022), the two objectives are in contrast with each other: most informative samples lie in the proximity of the decision boundary, thus tend to exhibit higher loss value (Aljundi et al. (2019); Buzzega et al. (2020b); Bang et al. (2021)); instead, methods that use the *memorization effect* select samples with smaller loss (Jiang et al. (2018); Han et al. (2018); Li et al. (2020)).

Remarkably, despite the opposite objectives, both benefit from AER; indeed, noisy and complex/rare samples tend to be forgotten faster than easy and clean samples. In light of this, we propose an *asymmetric* sampling objective, where for each $\mathbf{x} \in \mathcal{M}$ we define a *score* function $s(\mathbf{x})$:

$$s(\mathbf{x}) = \begin{cases} \mathcal{L}(\mathbf{x}, \tilde{y}), & \text{if } (\mathbf{x}, \tilde{y}) \sim \mathcal{D}_t \\ -\mathcal{L}(\mathbf{x}, \tilde{y}), & \text{if } (\mathbf{x}, \tilde{y}) \sim \mathcal{D}_{<t} \end{cases} \tag{4}$$

Then, we use $s$ to select the item to be replaced by prioritizing the release of those with **higher score**.

If we assume that the score of noisy samples far surpasses that of complex/easy ones, such a strategy ensures that most samples from the present are clean *without disregarding complexity*. Such an observation is supported both by our own empirical evidence (Sec. 5.1) and by the mathematical analysis of Maini et al. (2022). Hence, we switch the objective for samples from the past to only retain the most complex samples, *i.e.* those that have a stronger effect *against* forgetting. A depiction of this strategy can be seen in Fig. 2.

**Asymmetric Balanced Sampling (ABS)**. The score $s$ does not trivially allow the definition of a single probability distribution $p(\mathbf{x})$ from which to draw the sample to replace. For example, since $\mathcal{L}(\mathbf{x}, y) \geq 0, \forall (\mathbf{x}, y)$, simply normalizing $s(\mathbf{x})$ would strongly favour samples from either the current or the past task (more details can be found in the supplementary material). Instead, we ensure a balanced occurrence of elements from both past and present tasks. Let $\phi$ be a binomial distribution with probability $q = |\mathcal{M}_{curr}|/|\mathcal{M}|$:

$$p(\mathbf{x}) = \phi p_{curr}(\mathbf{x}) + (1 - \phi) p_{past}(\mathbf{x}), \tag{5}$$

where $p_{curr}(\mathbf{x}) = \sum \mathbf{x} \in \mathcal{M}_{curr}$ is the normalized score $s(\mathbf{x})$ for samples from the current task (vice-versa for $p_{past}$ using $\mathcal{M}_{past} = \mathcal{M} - \mathcal{M}_{curr}$). In Eq. 5, we use $\phi$ as a binary selector to choose whether to replace a sample from the present or the past. A detailed *algorithmic* overview of the entire procedure can be found in the supplementary material.

### 3.4 BUFFER CONSOLIDATION AND MIXMATCH

By combining AER with ABS we obtain a balance between purity – for samples of the current task – while preserving the complexity of those from the past. To achieve this, the backbone network had to be trained on a stream of noisy data. While we find that the effect of noise from the current task is mitigated by AER (Sec. 5.4), we can further reduce its influence with the help of the memory buffer.

In principle, with an ideal sample selection strategy we could simply train on samples from $\mathcal{M}$ to adjust the predictions of the network at the end of the task in a fully-supervised fashion (**buffer fit.**). While we empirically find in Sec. 4 that such a strategy delivers remarkable results, we can refine it to handle more complex noise scenarios.

In particular, we use a modified version of MixMatch (Berthelot et al. (2019)) to obtain a more robust model, using the most *uncertain* samples as a source for unlabeled data. Similarly to Arazo et al. (2019), we fit a two-component Gaussian Mixture Model (GMM) $g(\mathcal{L})$ on the loss $\mathcal{L}$ of each $(\mathbf{x}, \tilde{y}) \in \mathcal{M}$. Then, we compute the perceived uncertainty of each sample $u(\mathbf{x})$ as the posterior $g(l|\mathcal{L})$, where $l$ indicates the Gaussian component with a smaller mean. Samples are then separated into *pure* $\mathcal{P}$ and *uncertain* $\mathcal{U}$ with a simple threshold on $g(l|\mathcal{L})$.

From this, samples in $\mathcal{P}$ have label $\tilde{y} \approx y$, thus we can use them to compute a supervised loss term. Instead, for $\mathbf{x} \in \mathcal{U}$ we compute $\hat{y}$ using the model's response on different augmentations $T$ of $\mathbf{x}$:

$$\hat{y} = u(\mathbf{x})\tilde{y} + \frac{1 - u(\mathbf{x})}{\eta} \sum_{i=1}^{\eta} f_\theta(T(\mathbf{x})), \tag{6}$$

Table 1: Final Average Accuracy (FAA) [↑] across all tasks of different CNL methods on multiple datasets with different noise rates. [†]Additional baselines created by adapting existing loss-based and CL approaches to the multi-epoch scenario.

| Benchmark | Seq. CIFAR-10 | | | Seq. CIFAR-100 | | | | |
|---|---|---|---|---|---|---|---|---|
| | | *symm* | | | *symm* | | *asymm* | |
| Noise rate | 20 | 40 | 60 | 20 | 40 | 60 | 20 | 40 |
| Joint | 79.65 | 73.12 | 60.53 | 54.77 | 38.46 | 23.36 | 56.70 | 42.61 |
| Finetune | 18.83 | 18.01 | 15.99 | 08.65 | 07.55 | 06.15 | 07.78 | 05.73 |
| Reservoir (Vitter (1985)) | 50.53 | 33.64 | 22.92 | 25.14 | 14.64 | 8.92 | 29.42 | 18.91 |
| + *CoTeaching* (Han et al. (2018)) | 50.11 | 34.89 | 22.98 | 25.79 | 14.46 | 8.92 | 32.18 | 20.76 |
| + *DivideMix* (Li et al. (2020)) | 55.69 | 38.87 | 26.13 | 33.31 | 22.91 | 13.58 | 36.98 | 26.10 |
| GDumb (Prabhu et al. (2020)) | 35.45 | 27.76 | 19.41 | 16.96 | 11.31 | 7.62 | 17.25 | 11.75 |
| + *CoTeaching* (Han et al. (2018)) | 36.94 | 31.26 | 19.75 | 17.02 | 13.17 | 8.17 | 17.07 | 12.05 |
| + *DivideMix* (Li et al. (2020)) | 38.60 | 32.25 | 21.06 | 19.26 | 15.67 | 10.51 | 18.80 | 13.29 |
| PuriDivER (Bang et al. (2022)) | 30.96 | 27.23 | 24.31 | 27.53 | 24.36 | 17.81 | 25.46 | 18.84 |
| PuriDivER.ME[†] | 55.49 | 49.44 | 41.74 | 41.25 | 37.61 | 27.18 | 41.65 | 30.22 |
| DividERMix[†] | 57.07 | 45.65 | 32.19 | 29.21 | 22.41 | 14.21 | 29.38 | 21.23 |
| **OURs** | 60.82 | 59.47 | 45.07 | 44.34 | 38.64 | 26.34 | 41.24 | 29.26 |
| *w. buffer fit.* | **69.12** | 64.81 | 50.04 | **47.58** | **41.58** | 30.13 | 42.85 | 31.49 |
| *w. consolidation* | 68.82 | **67.14** | **54.59** | 46.11 | 40.27 | **34.81** | **43.67** | **32.64** |

Finally, we obtain the refined set $\mathcal{R} = \{(\mathbf{x}, \hat{y}) : (\mathbf{x}, \tilde{y}) \in \mathcal{U}\}$ and follow up with the MixMatch procedure to compute the supervised and self-supervised loss terms $\mathcal{L}_s$ and $\mathcal{L}_u$ respectively. The overall loss term is computed as $\mathcal{L}_s + \lambda_u \mathcal{L}_u$, where $\lambda_u$ is a regularization hyperparameter.

## 4 EXPERIMENTS

In line with other notable CL works (Rebuffi et al. (2017); Hou et al. (2019); Wu et al. (2019); Buzzega et al. (2020a); Bang et al. (2021); Boschini et al. (2022)), we adhere to a **multi-epoch setting**, in which samples can be experienced multiple times within the respective task. Additional results including an assessment of our implementation of PuriDivER, the applicability of PuriDivER's consolidation on our proposal, a deeper analysis of the computational costs, and a thorough ablative study on the impact of each component can be found in the supplementary material.

### 4.1 SETTING

We conducted experiments on five distinct datasets and explored various levels of noise to comprehensively assess the efficacy of our proposal across a diverse spectrum of tasks, including non-image-based classification. We use **CIFAR-10** and **CIFAR-100** datasets (Krizhevsky et al. (2009)), containing $32 \times 32$ colour images, and the **NTU RGB+D** (Shahroudy et al. (2016)) dataset for 3D skeleton-based human action recognition. On these datasets, we inject two types of synthetic noise commonly employed in literature (Li et al. (2020); Jiang et al. (2018); Han et al. (2018)) to replicate noisy labels: *symmetric* and *asymmetric* noise[*]. To address real-world label noise, we evaluate our method on **WebVision** (Li et al. (2017)) and on **Food-101N** (Lee et al. (2018)), composed of images gathered from the web, thus containing *instance-level* annotation noise.

We define sequential CL tasks for each dataset, under the ClassIL setting. Namely, for Seq. CIFAR-10 and Seq. WebVision and Seq. Food-101N, we split the classes into 5 tasks, while we consider a longer sequence of 10 tasks for Seq. CIFAR-100. For Seq. CIFAR-10/100, we employed ResNet18 (He et al. (2016)) as the backbone and conducted training on each task for 50 epochs. In the case of Seq. Food-101N and Seq. WebVision, ResNet34 (He et al. (2016)) was used as the backbone, with training for 20 epochs for the former and 30 epochs for the latter. For Seq. NTU-60, EfficientGCN-B0 (Song et al. (2021)) served as the backbone, and training was performed for 30 epochs.

---

[*]Additional details regarding the noise injection process can be found in the supplementary material.

Table 2: Final Average Accuracy (FAA) [↑] on Seq. NTU-60 and Seq. WebVision. † Additional baselines created by adapting existing loss-based and CL approaches to the multi-epoch scenario.

| Benchmark | Seq. NTU-60 | | Seq. WV |
|---|---|---|---|
| *noise rate* | 20 | 40 | N/A |
| Joint | 68.26 | 63.02 | 54.80 |
| SGD | 14.30 | 12.48 | 15.96 |
| Reservoir | 35.16 | 16.21 | 27.10 |
| *+ CoTeaching* | 44.43 | 32.03 | 27.80 |
| *+ DivideMix* | 40.92 | 32.07 | 29.93 |
| GDumb | 11.34 | 7.34 | 25.00 |
| *+ CoTeaching* | 13.81 | 9.18 | 25.20 |
| *+ DivideMix* | 15.96 | 6.59 | 28.00 |
| PuriDivER | 39.33 | 38.86 | 29.10 |
| PuriDivER.ME† | 43.10 | 38.07 | 36.40 |
| DividERMix† | 32.61 | 20.23 | 36.20 |
| **OURs** | 46.69 | 44.56 | 34.20 |
| *w. buffer fit.* | **49.59** | **48.18** | 36.84 |
| *w. consolidation* | 48.73 | 45.19 | **38.87** |

Table 3: Final Average Accuracy (FAA) [↑] of our method and main competitor on a real-world noisy dataset.

| Benchmark | Food-101N |
|---|---|
| Joint | 39.91 |
| **PuriDivER.ME†** | 29.23 |
| **OURs** | 29.86 |
| *w. buffer fit.* | **34.63** |

Table 4: Accuracy (FAA) comparison with SPR and CNLL. ‡ training iterations spread across epochs.

| Seq. CIFAR-10 – 40% *symm* | | | |
|---|---|---|---|
| **Buffer size (total)** | | 2500 | *unlimited* |
| CNLL | *1 epoch* | 38.14 | 57.26 |
| CNLL | *50 epochs* | 35.46 | 43.43 |
| OURs | *50 epochs* | **67.10** | **76.83** |
| **Buffer size (total)** | | 1000 | |
| SPR‡ | *25 epochs* | | 26.34 |
| OURs | *25 epochs* | | **63.65** |

The evaluation results are presented in terms of Final Average Accuracy (FAA), computed at the end of the final task as the average accuracy on all tasks and averaged across 5 runs. Due to space constraints, we refer the reader to the supplementary material for the results in terms of Final Forgetting, a detailed overview of our experimental settings, hyperparameters, and standard errors.

## 4.2 BASELINE METHODS

To assess the merits of our proposal, we compare it against PuriDivER, the current state-of-the-art sample selection strategy for CLN, as well as common rehearsal CL baselines adapted to our setting. For the latter, we follow Bang et al. (2022) and apply both CoTeaching and DivideMix to consolidate the buffer produced by ER and GDumb.

Since current CLN methods are designed for the online setting, a direct comparison may result in an unfair disadvantage and a weaker evaluation. Therefore, based on the considerations outlined in Sec. 3.2, we refine PuriDivER by suspending memory updates after the first training epoch; we name such method **PuriDivER.ME**. We also compare against SPR and CNLL, adapting the former for offline CLN and using the same overall memory budget for a fair comparison. Additional details regarding such methods and their adaptation can be found in the supplementary material.

In a similar fashion, we design an additional baseline by applying DivideMix on samples from both the current task and the buffer. Notably, this new method, which we name **DividERMix**, does not rely merely on buffer consolidation and can exploit all the data from the incoming data stream as a source of regularization. Finally, we compare against a model jointly trained on all tasks (*Joint*) and a lower bound derived by training without any countermeasure to forgetting or noise (*Finetune*).

## 4.3 COMPARISON WITH CNL BASELINES

Results for our main evaluation are reported in Tab. 1, 2 and 3‡. When multiple training epochs are allowed, methods that rely only on buffer consolidation become less effective, especially as the amount of noise increases. This is particularly true for GDumb: as it relies solely on the buffer, it cannot take advantage of the abundance of data along the task, and its performance is particularly limited. This is reflected by the performance of Reservoir-based baselines, as with low noise they considerably outperform GDumb in all our settings.

---

‡Due to spatial constraints we only report the comparison between our method and the second best.

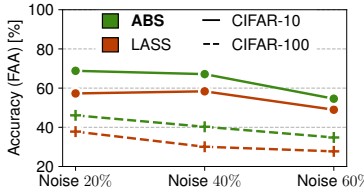

Figure 3: FAA ($[\uparrow]$) with sample selection performed by ABS and LASS.

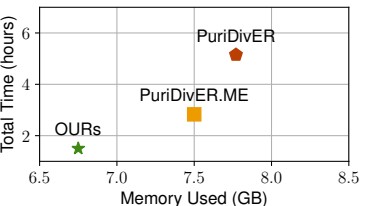

Figure 4: Training time $[\downarrow]$ and memory consumption $[\downarrow]$ for different methods.

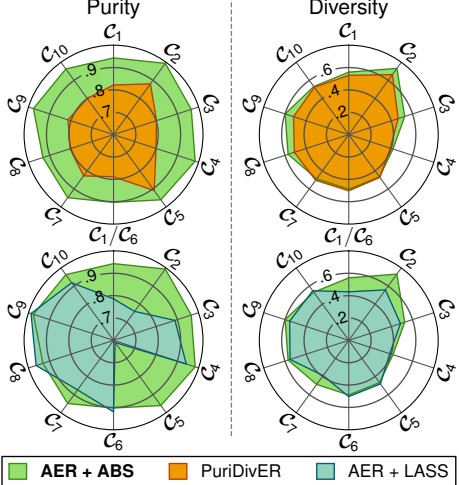

Figure 5: Final composition of the buffer with different choices of sample selection (best in color).

Such an outcome outlines the **potential benefits** of performing **multiple training iterations**, thus justifying the validity of our experimental setting. However, this comes as a double-edged sword, as we find a severe performance drop for all Reservoir-based methods as the stream becomes noisier.

As for SPR and CNLL, their significantly higher computational requirements make them inapplicable in all but the easiest CIFAR-10 dataset (Tab. 4), with sub-optimal performance due to degraded loss gaps. Remarkably, this remains true even in an unrestricted setting with relaxed memory constraints.

## 4.4 COMPARISON WITH MULTI-EPOCH CNL METHODS

We now turn our comparison to methods designed for the multi-epoch setting. First of all, we find an impressive performance gain for PuriDivER.ME w.r.t. PuriDivER ($15.04\%$ on average), which further supports our claims of Sec. 3.2. Both versions comprehend PuriDivER's consolidation Bang et al. (2022), with the model being fitted to buffer samples at the end of each CL task. Similarly, when compared with the DivideMix-based buffer consolidation, we find that our baseline DividERMix provides a performance gain in most scenarios.

However, both PuriDivER.ME and DividERMix are consistently surpassed by our proposal. In particular, we measure an average $2.25\%$ gain over the best competitor's performance **without any buffer consolidation**. On top of that, we find that a simple consolidation based on a fully-supervised optimization of the buffer – *buffer fit.* – provides an additional significant improvement ($3.78\%$ on average). This suggests that our proposal **successfully improves** the **purity and diversity** of samples in the buffer to the extent that it may be exploited without the need for extra regularization. However, as the sample selection is not perfect, under more complex noise scenarios our buffer consolidation based on MixMatch tends to prevail, with an average improvement of $7.47\%$ w.r.t. the baselines. Finally, these considerable gains come with a remarkable speed-up in terms of both time and resources used (Fig. 4), making it more suitable for a multi-epoch incremental scenario.

## 5 MODEL ANALYSIS

### 5.1 PURITY OF THE BUFFER

Our main desiderata is to achieve a balanced set of clean samples while retaining the most relevant samples for later replay. Thus, in Fig. 5 we evaluate the effective *purity* and *diversity* of $\mathcal{M}$ after training on Seq. CIFAR-10 with $40\%$ noise. For each class $\mathcal{C}_i$, we define purity as $\mathbb{E}_{y \in \mathcal{M}[\mathcal{C}_i]}[\mathbb{1}_{\tilde{y}=y}]$, while for diversity we estimate the intra-class variation as the average standard deviation of the features produced by the *Joint* ideal model. Finally, we account for the attained balancing between the different classes by scaling all values by the *prior probability* of the respective class in $\mathcal{M}$.

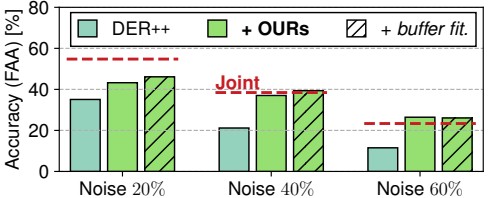
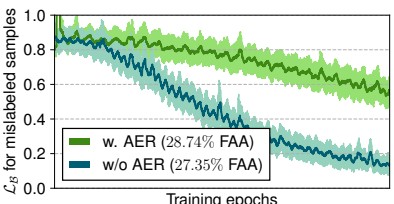

Figure 6: FAA ([↑]) of DER++ with our method and buffer fitting.

Figure 7: Effect of AER on the speed at which the model learns the noisy data.

We compare ABS against PuriDivER.ME and a simple Loss-Aware Symmetric Sampling (**LASS**), where each sample is assigned a score proportional to its loss. As shown in Fig. 5, ABS clearly outperforms both LASS and PuriDivER.ME in terms of purity and diversity. Unexpectedly, LASS results in a particularly unbalanced buffer, with only the last few classes presenting a good balance; this translates into a significant performance drop of around $8.59\%$ on average (Fig. 3). Instead, while PuriDivER.ME is better at balancing the buffer, it falls behind in terms of purity.

## 5.2 STABILITY AND GENERALIZATION CAPABILITY

We investigate the effect of noisy data in the memory buffer in terms of *flatness* of the loss landscape. Previous works have defined the smoothness of the decision boundaries as a key factor for generalization (Keskar et al. (2017); Neyshabur et al. (2017)), while in CL the stability of the attained local minima has been shown to lead to a lower degree of forgetting (Buzzega et al. (2020a); Boschini et al. (2022)). As in Bonicelli et al. (2022), we measure the Lipschitz constant $L$ of the model, as a lower value has been linked to stronger robustness against perturbations (Cisse et al. (2017); Leino et al. (2021)) and smoother decision boundaries (Xu & Mannor (2012); Szegedy et al. (2014)).

## 5.3 APPLICABILITY TO OTHER CL METHODS

In this section, we aim to prove that our method remains effective regardless of the specific underlying rehearsal method employed. We do so by applying AER and ABS to **DER++**. Here, to estimate $s(\mathbf{x})$ we measure the overall replay loss of the method, consisting of a distillation term and the replay regularization term of Eq. 3. In addition, we include the fully-supervised finetuning on the buffer (see Sec. 4.4), to observe its impact. The results in Fig. 6 depict that our proposals successfully enhance the capabilities of other CL baselines, thus substantiating our initial claims.

## 5.4 EFFECTIVENESS OF AER AS A REGULARIZER FOR CNL

In Fig. 7, we depict the effects of *enabling and disabling AER* during the second task of Seq. CIFAR-10 on the loss of noisy samples from the current task. Surprisingly, we find that *AER vastly reduces the rate of convergence of noisy samples*, which just by itself improves over the baseline in terms of FAA. Indeed, providing a purified and diverse set of examples to counter forgetting is only part of the challenge: as the model is subjected to a continuous stream of noisy data from the current task, it becomes important to also reduce the speed with which noisy samples from the present are learned.

## 6 CONCLUSIONS

Our study presents an innovative framework for addressing the challenge of Continual Learning in the presence of Noisy Labels, a common issue in real-world AI applications. Focusing on the established multi-epoch class-incremental scenario, we find that current methods using the small-loss criterion fall short: as training ensues, the loss gap between clean and mislabeled samples collapses. To overcome this limitation, we appeal to a long-standing enemy of continual learning – *forgetting* – and propose Alternate Experience Replay to maintain a clear separation between mislabeled, complex, and clean samples. We also introduce Asymmetric Balanced Sampling to enhance sample diversity and purity within the buffer. Through extensive experiments, our approach outperforms competitors by a wide margin, showcasing its potential to significantly improve learning in noisy label scenarios.

## ETHICS STATEMENT

We believe that this work does not possess any harmful applications that could negatively affect the public. However, it is important to note that our approach, which relies on rehearsal techniques, involves the storage of raw data and may not be suitable for scenarios where privacy constraints are of utmost importance.

## REPRODUCIBILITY STATEMENT

We provide full code implementation and hyperparameters for all the algorithms covered by our evaluation as part of the supplementary material. Notably, AER and ABS do not involve additional hyperparameters, with only the learning rate being tuned.

Our evaluation involves the injection of random noise to simulate an imperfect process of annotation. To allow the reproduction of our results, we include as part of the code the newly generated annotations for all datasets and noise scenarios.

In compliance with the Terms and Conditions of the NTU RGB-D dataset, we do not provide links to download the aforementioned dataset.

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
