# OpenReview forum: "May the Forgetting Be with You: Alternate Replay for Learning with Noisy Labels"
_ICLR.cc/2024/Conference — Submitted to ICLR 2024_

### Official Review · Reviewer_QKiJ · 2023-10-23

**Soundness:** 2 fair
**Presentation:** 2 fair
**Contribution:** 1 poor
**Rating:** 3
**Confidence:** 4

**Summary:**

In this study, the authors leverage the phenomenon of forgetting to tackle a specific challenge: continual learning under noisy labels. They posit that samples affected by noisy labels are more prone to being forgotten, a characteristic that can be exploited to sift clean samples from noisey datasets. To achieve this, they introduce Alternate Experience Replay (AER), a mechanism that select potential clean samples in the current task while concurrently replaying samples from previous tasks. The efficacy of their proposed method is assessed using both synthetic and real-world datasets.

**Strengths:**

1. Combining continual learning with label noise is interesting, although this setting is controversial.
2. The paper is well-structured and easy to follow.

**Weaknesses:**

1. The techniques proposed in the study, specifically 'forgetting' in the realm of learning with noisy labels and 'replay' within continual learning, aren't novel in their individual contexts. The mere combination of these two established methods doesn't inherently bring novelty to the field.
2. The authors advocate the use of forgetting as a selection criterion, yet its effectiveness may be limited to specific types of label noise. It appears inadequate for more complex noise categories, for instance, pairflip-45 and instance label noise, let alone for real-world datasets. A more extensive experimental analysis is essential to thoroughly investigate the 'forgetting' phenomenon's applicability and limitations in various noise scenarios.
3. The experiments are confined to small datasets, with only 10 classes from WebVision utilized, which casts doubt on the scalability of the proposed methods. For a dataset of 50k, joint learning seems more appropriate, and the unimpressive outcomes in Table 1 underscore the proposed methods' shortcomings. Similarly, the results in Table 2 are too inconclusive for any solid verification. The authors should report the results of other label noise methods such as DivideMix on Joint.

**Questions:**

1. Why are the results of CIFAR-10 with asymmetric label noise missing in Table 1?

---

> ### Author Response · Authors · 2023-11-17
> **Real world experiments and setting choices**
>
> Following the reviewers' suggestion we conduct additional analysis on the *real-world* noisy dataset, Food-101N, as done in [PuriDivER]. The results show that our method produces an increase of $4.52$% accuracy points w.r.t. our second-best PuriDivER.ME and of $5.40$% accuracy points when 'fine-tuning on the buffer' is enabled for both methods,  with our method being only $5.28$% accuracy point under the upper-bound (Joint).
>
> | **Benchmark**         | **Food-101N** |
> |------------------------|---------------|
> | Joint                  | $39.91$       |
> | **PuriDivER.ME**       | $25.34$       |
> |   *w. buffer fit.*     | $29.23$       |
> | **OURs**               | $29.86$       |
> |   *w. buffer fit.*     | **$34.63$**   |
>
>  We also note that our work comprises an initial empirical evaluation of the behaviour of clean and noisy data in the continual learning scenario, out of which the method is then built. On this premise, and in line with the current literature [PudiDivER,SPR], instance-independent label noise allows us to carry out such analyses. Moreover, we want to stress that we closely follow the settings of the other CNL methods and therefore conduct experiments suitable for comparisons. For what concerns the other types of noise, we firstly want to remark that the **asymmetric** noise setting, also called *class-dependent* label noise, is indeed an approximation of real-world corruption patterns. Specifically, in the CIFAR-100 dataset, images are grouped based on their relative superclasses, thus each image comes with a "fine" label (specific class) and a "coarse" label (superclass). The asymmetric noise injection alters labels within the same superclass. This results in sample ambiguity occurring only between similar classes as it would in a **realistic** scenario. Secondly, instance-dependent noise has not yet been extensively investigated and only a few studies address it, owing to its complex modelling [ALNL].
>
> We are unsure about what makes the reviewer refer to our results as *unimpressive*, whereas the other reviewers valued our work for the *novelty*, the *comprehensive experiments* (**2XeP**), the *effectiveness* (**Jj1M**) of the method and the *thoroughness* of the analysis (**aFAe**).
>
> As for the **question** on the CIFAR-10 experiments missing: due to space constraints, we prioritize more complex and longer benchmarks.
>
>     [PuriDivER]: Bang, Jihwan, et al. "Online continual learning on a contaminated data stream with blurry task boundaries." In CVPR. 2022.
>     [SPR]: Kim, Chris Dongjoo, et al. "Continual Learning on Noisy Data Streams via Self-Purified Replay." In ICCV. 2021.
>     [ALNL]: Song H, Kim M, Park D, Shin Y, Lee JG. Learning from noisy labels with deep neural networks: A survey. IEEE Transactions

---

### Official Review · Reviewer_Jj1M · 2023-10-30

**Soundness:** 2 fair
**Presentation:** 3 good
**Contribution:** 2 fair
**Rating:** 5
**Confidence:** 4

**Summary:**

This paper tries to solve the continual learning problem under noisy labels by proposing Alternate Experience Replay (AER). The author found that the loss of clean data and noisy data have different trends that can help the model find which data has noisy labels. From this perspective, the author proposed Asymmetric Balanced Sampling to improve the AER performance.

**Strengths:**

+ The proposed method is well-motivated and easy to understand.
+ The experiments demonstrate the effectiveness of this method.

**Weaknesses:**

+ I believe the alternate replay is an interesting design, but the novelty of the proposed method is still limited. The strategies used in this paper are simple and very common techniques, such as selecting clean samples based on loss thresholds and MixMatch. It seems like a direct combination of existing technologies, with no essential innovation in methodology, especially for continual learning.
+ It is difficult to adapt to more complex noise distribution by selecting samples directly based on the loss threshold. More advanced methods should be considered [1, 2] instead of still adjusting the threshold manually, which is not helpful for either CL or noisy label problems.
+ PuriDivER is designed to handle online CL and blurry tasks with noisy labels. Therefore, it may not be suitable for offline CL. However, both online CL and blurry tasks are more challenging and realistic settings, and I’m curious how well the proposed method AER works under these settings as well. In particular, how to adapt AER to online CL where the distribution changes rapidly?
+ The comparison methods cannot demonstrate the real effectiveness of AER and ABS. Since few methods focus on offline CL with noisy labels, the authors should conduct more detailed comparative experiments. For example, using the same ABS strategy, compare the performance of AER and DividMix to verify the excellence of AER. In addition, using the same AER strategy, the performance of ABS and PuriDivER sampling methods should also be compared.
   ```
   [1] Meta label correction for noisy label learning. AAAI-2021.
   [2] Learning from noisy labels with decoupled meta label purifier. CVPR-2023.
   ```

**Questions:**

Please refer to the weakness section.

---

> ### Author Response · Authors · 2023-11-18
> **Comments on weaknesses 1 and 2**
>
> **Weakness 1**:
> We struggle to understand the point raised by the reviewer; they recognize the novelty of Alternate Replay (the main contribution of our work) but still find the contribution not good enough.
>
> As the vast majority of current LNL works, our work combines existing techniques to bring new ideas and methodologies (i.e. our **AER** and **ABS**). While we borrow and draw inspiration from well-established techniques in the literature about noisy labels, they do not constitute our primary contributions.
>
> Indeed, as appreciated by reviewer 2XeP, our work rearranges experience replay to handle noisy label problems, introducing a novel bi-fold sampling strategy which favours the presence of clean data from both the past and current tasks. Such a strategy exploits the fact that mislabeled examples are forgotten faster than regular ones and introduces buffer forgetting to separate clean and noisy in-memory samples.
>
> Notably, we are the first to address incremental learning under the *noisy* and *offline* setup, whereas the current existing works focus on the online setup. For example, PuriDivER cannot adequately fit complex distributions when only one epoch is allowed, as holds for FOOD-101N.
>
>  **Weakness 2**:
>
> The methods suggested by the reviewer rely on meta-learning to learn a label correction mechanism. To do so, these approaches need a small *clean validation set*.
> We argue that such a requirement might clash with our setting, tailored for realistic scenarios. Indeed, several real-world noisy datasets do not come with annotations regarding which examples are "clean" and which are not.
> Hence, we believe that the direct application of [1],[2] is slightly out-of-scope for our work; we will modify our manuscript to include the suggested works in the related materials.
>
> Regarding the simplicity of our sample selection technique, we have a different perspective and see it as a strength of our approach.
> We emphasise that our algorithm has a single hyper-parameter (the learning rate) and does not require any other tuning (we use the same threshold for each dataset).
>
> Nonetheless, to assess the merits/validity of our choice, we conducted new experiments by fitting the more "complex" Gaussian Mixture Model (GMM) on per-sample loss values to divide training samples into labelled *vs* unlabeled sets.
> We equip our approach with this selection technique and report here a comparison with our original "simple" approach (we do so for both synthetic - CIFAR-100 - and real label noise - FOOD-101N - datasets).  For the experiments with synthetic noise, since we have information regarding which targets are noisy or not, we can report the buffer cleanliness percentages at the end of training for both selection techniques.
>
> **Final Average Accuracy**
> | Benchmark   | Seq. CIFAR-100 | Seq. CIFAR-100 | Seq. FOOD-101N	|
> |-------------|----------------|----------------|----------------|
> | Noise rate  |  *asymm*  20            | *asymm* 40             | *real-world* |
> | **OURs**  | **$41.24\pm0.40$** | **$29.2\pm0.91$** | $29.86\pm0.35$ |
> | OURs w. **GMM** sample selection   | $39.25\pm0.60$    | $26.8\pm0.90$      | $23.41\pm0.47$|
>
>
> **Final percentage of clean samples in the buffer**
> | Benchmark   | Seq. CIFAR-100 | Seq. CIFAR-100 |
> |-------------|----------------|----------------|
> | Noise rate  | *asymm*   20            |  *asymm* 40             |
> | **OURs**  | **0.74** | **0.96** |
> | OURs w. **GMM** sample selection   | 0.66   |  0.89  |
>
> With the results in the tables above, we show that the simple sample selection technique adopted proves more effective at separating clean samples. Nevertheless, sample selection via GMM remains a valuable option to avoid additional hyperparameters.

---

> ### Author Response · Authors · 2023-11-18
> **Comments on weaknesses 3 and 4**
>
> **Weakness 3**:
> Despite the reviewer considering the offline setting unrealistic and negligibly important, we would like to clarify that our experimental setting aligns with many notable works [PASS, DER, LIP, ICARL, LUCIR, CO2L, LSIL, PODNET].
> In this respect, we advocate for what has been said in [DER]: when only one epoch is allowed on the current task, even the SGD baseline fails at fitting it with adequate accuracy, especially with complex datasets such as CIFAR-100 and FOOD-101N. Therefore, the resulting performance - and in turn the comparisons among different approaches - can be difficult to read, as the effects of catastrophic forgetting and those linked to underfitting interleave here.
>
> Since in our work Alternated Replay is activated/deactivated every other epoch, the method does not fit the online training scenario.
>
> [PASS]: Zhu et al. Prototype augmentation and self-supervision for incremental learning. CVPR 21.
> [DER]: Buzzega et al. Dark experience for general continual learning: a strong, simple baseline. aNeurIPS 20.
> [LIP]: Bonicelli et al. On the Effectiveness of Lipschitz-Driven Rehearsal in Continual Learning. aNeurIPS 21.
> [ICARL]: Rebuffi et al. icarl: Incremental classifier and representation learning. CVPR 17.
> [LUCIR]: Hou et al. Learning a unified classifier incrementally via rebalancing. CVPR 19.
> [CO2L]: Cha et al. Co2L: Cooperative continual learning for lifelong visual recognition. ICCV 21.
> [LSIL]: Wu et al. Large Scale Incremental Learning CVPR 19.
> [PODNET]: Douillard et al. 8: Pooled Outputs Distillation for Small-Tasks Incremental Learning ECCV 20.
>
> **Weakness 4**:
> We here report some comparative experiments on the contribution of our AER and ABS on the two methods cited by the reviewer (for both CIFAR-100 and NTU-60 datasets).
>
> We observe from the following tables that for both DivideERMix and PuriDivER, the addition of AER and ABS has always a positive impact on performance. The increase is more noticeable in PuriDivER than in DividERmix, thus supporting our choice of having it as our main competitor.
>
> | **Benchmark**          |  **Seq. CIFAR-100**     | **Seq. CIFAR-100** |    **Seq. CIFAR-100**   |
> |------------------------|-----|---------------------|-----|
> | **Noise rate (symm)**  | 20% | 40%                 | 60% |
> | **OURs** *w. AER w. ABS* | $47.58$ | $41.58$ | $30.13$ |
> | **DividERMix** | $29.21$ | $22.41$ | $14.21$ |
> |   *w. AER* | $32.63$ | $24.52$ | $14.35$ |
> |   *w. ABS* | $32.95$ | $25.35$ | $14.43$ |
> | **PuriDivER** | $27.53$ | $24.36$ | $17.81$ |
> |   *w. AER* | $28.90$ | $24.68$ | $23.00$ |
>
> | **Benchmark**                  | **Seq. NTU-60** | **Seq. NTU-60** |
> |--------------------------------|-|-|
> | **Noise rate (symm)**          | 20% | 40% |
> |  **OURs** *w. AER w. ABS* | $49.59$ | $48.18$ |
> | **DividERMix**                | $32.61$ | $20.23$ |
> |   *w. AER*                    | $32.98$ | $26.39$ |
> |   *w. ABS*                    | $38.49$ | $24.92$ |
> | **PuriDivER**              | $39.33$ | $38.86$ |
> |   *w. AER*                    | $47.39$ | $39.56$ |

---

> ### Comment · Reviewer_Jj1M · 2023-11-22
> **Thanks for the authors' response**
>
> I appreciate the authors for their response. I would like to keep my score unchanged. At least for now, I don't think it's necessary to consider noisy label settings in the current continual learning (CL), since CL itself remains many challenges such that its performance is quite poor compared to the offline learning manner. In my personal opinion, we should guide ones in this field to explore that truly improves continual learning rather than considering some very special settings.

---

> ### Author Response · Authors · 2023-11-22
>
> We thank the reviewer for taking the time to respond. We remark that, contrarily to the reviewer's beliefs, the current advances in CL have shown performance approaching that of the offline (joint) counterpart [l2p, coda-prompt, attriclip, slca, first\_session\_adaptation] at least in image classification tasks. Moreover, we observe that plenty of "special settings" have contributed significantly to the performance of current CL methods (e.g., CL and unsupervised representation learning [lump (NeurIPS 2021)] or domain adaptation [ci-uda (ECCV 2022)], CL under limited supervision [ccic (PRL 2022), nnscl (CVPR 2023 - oral)], neuro symbolic CL [cool (ICML 2023)], continual predictive learning [cpl (CVPR 2022 - oral)], CL and reinforcement learning [csp (ICLR 2023 - oral)], ...). Learning in the presence of noisy labels considers the non-ideal but highly plausible case in which the dataset contains annotation errors. Notably, this is the case of most real-world datasets and most of the literature benchmarks e.g., the Imagenet dataset has a label error rate of 5-6\% [label\_errors\_benchmarks].
>
> Finally, we would expect and solicit the reviewer to give a technical comment about the soundness, applicability, and whether the rebuttal has answered the questions that determined the original score. Notably, we were surprised about the reviewers' motivations for being so firm in their decision. Indeed, we feel that their judge is highly biased and not truly dependent on the content of the article and its technical/scientific contributions.
>
>     [label_errors_benchmarks]: Curtis Northcutt, et al. "Pervasive Label Errors in Test Sets Destabilize Machine Learning Benchmarks." NeurIPS 2021.
>     [l2p]: Wang, Zifeng, et al. "Learning to prompt for continual learning." CVPR 2022.
>     [coda-prompt]: Smith, James Seale, et al. "CODA-Prompt: COntinual Decomposed Attention-based Prompting for Rehearsal-Free Continual Learning." CVPR 2023.
>     [attriclip]: Wang, Runqi, et al. "AttriCLIP: A Non-Incremental Learner for Incremental Knowledge Learning." CVPR 2023.
>     [slca]: Zhang, G., Wang, L., Kang, G., Chen, L., & Wei, Y. (2023). SLCA: Slow Learner with Classifier Alignment for Continual Learning on a Pre-trained Model. ICCV 2023.
>     [first_session_adaptation]: Panos, Aristeidis, et al. "First Session Adaptation: A Strong Replay-Free Baseline for Class-Incremental Learning." ICCV 2023.
>     [nnscl]: Kang, Zhiqi, et al. "A soft nearest-neighbor framework for continual semi-supervised learning." CVPR 2023.
>     [ccic]: Boschini, Matteo, et al. "Continual semi-supervised learning through contrastive interpolation consistency." PRL 2022.
>     [lump]: Madaan, Divyam, et al. "Representational continuity for unsupervised continual learning." ICLR 2022.
>     [cool]: Marconato, Emanuele, et al. "Neuro symbolic continual learning: Knowledge, reasoning shortcuts and concept rehearsal." ICML 2023.
>     [ci-uda]: Lin, Hongbin, et al. "Prototype-guided continual adaptation for class-incremental unsupervised domain adaptation." ECCV 2022.
>     [cpl]: Chen, Geng, et al. "Continual predictive learning from videos." CVPR 2022.
>     [csp]: Gaya, Jean-Baptiste, et al. "Building a subspace of policies for scalable continual learning." ICLR 2023.

---

> ### Comment · Reviewer_Jj1M · 2023-11-23
>
> + The settings in the paper are far away from pretrain-based continual learning (CL) models. Additionally, the authors haven't discuss these methods in the paper, and there is no theoretical or experimental support for their work to be applicable to pretrain-based CL models.
> +  I would like to maintain my score just according to the authors' response, rather than what I previously stated regarding "At least for now, ..., we should guide ones in this field to explore that truly improves continual learning rather than considering some very special settings." I apologize to the author for any confusion caused.

---

> > ### Author Response · Authors · 2023-11-23
> >
> > From the reviewer's comment, we now find out that their ground for our rejection seems to be the fact that we did not cite pretrain-based CL models. In light of this, we believe that, despite our utmost efforts to answer the weaknesses raised, the reviewer remains committed to *avoid judging our work based on its technical or scientific novelty*, as they ignored any argument we made during the rebuttal.

---

### Official Review · Reviewer_aFAe · 2023-10-31

**Soundness:** 2 fair
**Presentation:** 3 good
**Contribution:** 2 fair
**Rating:** 6
**Confidence:** 4

**Summary:**

This paper explores the problem of continual learning under noisy labels. The authors mainly designed Alternative Experience Replay and Asymmetric Balanced Sampling to address the problem. At the same time, the authors also adopted consolidation and MixMatch to strengthen their algorithm. In the experiment, the authors validated their algorithms on 4 datasets and showed state-of-the-art results. They also performed a thorough analysis of the algorithm.

**Strengths:**

The authors perform a thorough analysis both in exploring the problem and in the effect of the algorithm. Each component of the algorithm is well motivated. The intuition of the algorithm is well presented.

**Weaknesses:**

1. The  main concern is about the performance of the algorithm.
- The authors conducted experiments on four datasets, three of which were synthesized, and one contained natural noise (WebVision). The discussions and improvements primarily focused on the synthesized datasets. Regarding the WebVision dataset, it remains unclear why the authors selected only 10 classes out of the available 1000 classes, and the criteria for selecting these 10 classes are also not clear. According to Table 2, the AER + ABS approach did not outperform PuriDivER. The proposed method marginally outperformed PuriDivER only with the implementation of consolidation and MixMatch, which were adopted from other papers. It is of great concern that the algorithm can only function effectively in the synthetic scenario.
- Ablation study of AER and ABS is needed.
- Is there explanation of PuriDivEr's consolidation?
- It is not clear whether the experiments are in offline continual learning or online continual learning. The single or multiple training iterations is decided by the offline/online setting, and not something that an algorithm could choose. (section 4.3)
- It would be beneficial if the authors also compared it to PuriDivER when applying the algorithm to DER++.
2. Some motivation and details of the algorithm are not clear.
- In Figure 1, the authors aim to demonstrate the change in loss of noisy data with and without replay to highlight that noisy data tends to exhibit greater forgetting. However, it remains unclear in the figure which task the noisy data originates from. Moreover, the depiction of forgetting would be clearer by comparing the accuracy (loss) of the same data from different tasks, rather than comparing the same task with different data or comparing the same data at different stages.
- It is not clear if Eq(1) can represent the optimization goal. If I understand correctly, Eq(1) means finding a model to fit the noisy distribution, even when given a wrong label. The loss function yields a smaller value when the model also predicts the wrong label. However, the algorithm seems to eliminate the influence of the noisy label and encourages the model to predict the correct label.
- In Eq(4), is the score function $s(x)$ for task $t$? In this case, why would the buffer contain the current task data?
- In the motivation of ABS, why $\mathcal{L} \geq 0$ cause $s(x)$ favour samples from the current task?
- In the ABS part, what is $p_{curr}$ and $p_{past}$?
3. There are also some writing issues, mainly related to logic, that make the paper harder to read at some important points.
- The logic is weak in the motivation part of the abstract -- noise scenarios (caused by limited annotation time) renders the CL with buffer vulnerable
- It is wired to mention "adaption is faster" (page 2 line 3) when explain CL is vulnerable under noise label.
- In section 3.4, line 2, "the backbone had to be trained on a stream of noisy data". This is exactly the setting. This sentence is confusing.

**Questions:**

See the weaknesses part.

---

> ### Author Response · Authors · 2023-11-16
> **Comments on Weakness 1**
>
> We appreciate the reviewer's suggestion and comments about both strenghts and weaknesses, inviting further discussion to enhance the paper and its evaluation.
> **Weakness 1**:
> * Similarly to the approaches in Continual Learning under Noisy Labels (CLN) studies [PuriDivER, SPR], we opt for selecting classes from the WebVision dataset by choosing *the ones containing the largest number of images each*. This decision is motivated by observations from the PuriDivER experiments on the WebVision and Food-101 datasets presented in the original paper, where the method noticeably underfits. Our emphasis is on addressing *forgetting* rather than the underfitting resulting from having numerous classes with varying cardinalities, given the continual learning scenario we are tackling. Following this and other comments from the reviewers, we have incorporated an experiment with **Food-101N**, and now we evaluate two datasets with *real-world noise* We would like to include these additional results in the main paper should the reviewers reconsider their ratings.
> | Benchmark   | Food-101N |
> |---------------------|-----------|
> | Joint   | $39.91$   |
> | PuriDivER.ME  | $25.34$   |
> |  w. buffer fit. | $29.23$   |
> | OURs    | $29.86$  |
> |  w. buffer fit. | **34.63** |
>    * [SPR]: Kim, Chris Dongjoo, et al. "Continual Learning on Noisy Data Streams via Self-Purified Replay." In ICCV. 2021.
>    [PuriDivER]: Bang, Jihwan, et al. "Online continual learning on a contaminated data stream with blurry task boundaries." In CVPR. 2022.
> * It is important to highlight that PuriDivER inherently includes the buffer fitting stage as a vital aspect of the method, while our AER+ABS does not. Consequently, a fair comparison should be drawn between our method, incorporating the buffer fitting stage, and theirs. It is noteworthy that we have meticulously tailored Puridiver.ME to our offline setting, with consistently superior results compared to PuriDivER. Additionally, we decided to also include the performance of PuriDivER.ME with the *buffer fitting stage disabled*. Results are reported in the following table and we argue these provide a fair comparison between our method and PuriDivER.ME.
> Furthermore, the results on the newly introduced real-world noisy dataset (Food-101N) validate the efficacy of our method even in the absence of buffer fitting.
>
> | Benchmark   | Seq. CIFAR-100 | Seq. CIFAR-100 |
> |-------------|----------------|----------------|
> | | symm  | asymm |
> | Noise rate | 60  | 40   |
> | OURs  | $\mathbf{28.60\pm0.55}$ | $\mathbf{31.32\pm1.35}$ |
> | PuriDivER.ME (w/o buff. fit)   | $15.56\pm0.46$ | $23.22\pm0.50$ |
>
> * As for the evaluations they consider lacking, these are precisely documented in the **supplementary materials** provided alongside the main paper. As evident from the results in Tab. E (reported again here), the introduction of each additional feature contributes to a performance improvement, particularly noteworthy when incorporating AER and ABS individually, as well as in combination.
>
>  Benchmark | Seq. CIFAR-100 | Seq. CIFAR-100 | Seq. CIFAR-100 | Seq. CIFAR-100 | |
> |-----------|----------------|----------------|----------------|----------------|----------------|
> | ER-ACE    | $\alpha$  | AER       | ABS       | **FAA**  |
> | &check;   |           |           |         | 11.65      |
> | &check;    | &check;    |           |       | 19.97      |
> | &check;   | &check;    | &check;   |      | 24.19      |
> | &check;   | &check;   | &check;   | &check;   | **26.34**  |
> | &check;   | &check;   |           | &check;   | 21.68      |
>
> * We would like to clarify that the consolidation phase of the Puridiver method is not part of our approach. A single experiment was presented in the supplementary material for the purpose of comparison, validating the robustness of our consolidation. Nonetheless, we use this term to denote the technique employed by both PuriDivER and PuriDivER.ME to distinguish between clean and noisy data, subsequently applying pseudo-labeling in a DivideMix fashion.
>
> * We concur with the reviewer's observation that "The single or multiple training iterations are decided by the offline/online setting, and not something that an algorithm could choose." The references to offline or online settings in the main paper are made to highlight that some competitors were tailored for online Continual Learning (CL), whereas our method is explicitly designed for offline CL. We will enhance the clarity on this aspect throughout the entire paper, with the hope that the reviewer will reconsider their evaluation.
>
> * Following the suggestion of the reviewer, we compare experiments already reported in the main paper of AER and ABS applied to DER++ with PuriDivER relying on DER++, which we name PuriDivDER++. Results are reported in the following table
>
> | Benchmark | Seq. CIFAR-100 |
> |-------------|----------------|
> | | symm|
> | Noise rate  | 20 |
> | DER++ | 35.09 |
> |  w.OURs | **43.25** |
> | PuriDivDER++| 35.58|

---

> > ### Author Response · Authors · 2023-11-16
> > **Comments on Weakness 2**
> >
> > **Weakness 2**:
> > * We wish to emphasize that the caption of the same figure already specifies that the data originates from the second task. Plotting the same figure at the first task, where the model exclusively replays data from the same task, would have demonstrated behaviours already well-documented and extensively shown in the literature [SSF,toneva2018ans]. Furthermore, our intention was to illustrate loss separation in the continual scenario and that with no countermeasures it becomes impossible to distinguish clean from noisy samples from the **current** task. Therefore, the second task provides the first favourable opportunity to demonstrate that loss separation between clean and noisy data also occurs in a continual learning scenario. We avoid comparing the same data from different tasks as each rehearsal process involves storing data in the buffer upon encountering samples from the current task. Therefore, with Fig. 1(b) we aim to demonstrate how we can leverage forgetting during the current task to more effectively distinguish between noisy and clean data. Overall, Fig. 1 serves to illustrate the intuition behind our AER.
> > * We appreciate the reviewer's suggestion about and build upon it to **re-arrange Sec. 3.1** such that the optimization goal is not to encourage the model to learn the wrong labels, which is not our desiderata.
> > * The *reservoir* mechanism employed in most rehearsal methods literature is crafted to produce an i.i.d. representation of the original distribution and insert data from the current task into the buffer while it is in progress. Consequently, we adhere to the same methodology.
> > *  In ABS, the scores are defined as
> > $ s(x) = L(x), \text{if}  \quad x \in D_t$
> > $\qquad \quad  -L(x) \text{if} \quad x \in D_{< t}$
> >
> > Such a score denotes the chance for a given example to be replaced. However, before sampling from $s$ we need to normalize the scores, thus we define a normalization factor $z$ = $\sum_{i} s(x_i)$, $\forall$ $x_i$ $\in$ M and compute the normalized scores of current and past task buffer samples as
> >
> > $ s(x) = p_{curr}(x) = \frac{L(x)}{z}, \text{if}  \quad x \in D_t$
> > $\qquad \quad  p_{past}(x) = -\frac{L(x)}{z} \text{if} \quad x \in D_{< t}$
> > thus, since  $\mathcal{L}(\mathbf{x}) \geq 0,  \forall \mathbf{x}$, it follows that $\frac{\mathcal{L}(\mathbf{x})}{z} \geq -\frac{\mathcal{L}(\mathbf{x})}{z}$ .
> >
> > We thank the reviewer for their concern and following their feedback we will include this detailed explanation in the final revision.
> >
> > [SSF]: Maini, Pratyush, et al. "Characterizing datapoints via second409 split forgetting." In aNeurIPS, 2022.
> > [toneva2018an]: Toneva, Mariya, et al. "An empirical study of example forgetting during deep neural network learning." In ICLR Workshop, 2019.

---

> > > ### Comment · Reviewer_aFAe · 2023-11-23
> > > **Response to the authors**
> > >
> > > Thank you so much for the response. I still have several concerns in the following
> > >
> > > 1. About the benchmark, the authors mentioned
> > > > Our emphasis is on addressing forgetting rather than the underfitting resulting from having numerous classes with varying cardinalities
> > >
> > > I do not think classification within more than 10 classes is a scenario outside of the goal of CL under noisy label. Furthermore, I did not find any statistics of Food 101-N descriped in the main paper or supplementary.
> > >
> > > 2. As for the buffer fit and consolidation, it sounds more like a robust baseline based on the author's description. Generally speaking for the presentation of a paper, if it relies heavily on existing methods, like PuriDivER in this case, it seems more logical to use these proven components as a baseline, and allows for the modification of specific aspects where improvements are possible, rather than conducting an ablation study with the existing methods. And if they should be presented as plug-able components, we need at least one sentence to describe them in the main paper. But I could not find the description for consolidation, and that's why I ask.
> > >
> > > 3. In Figure 1, if the loss curve exclusively represents samples from the second task(current task), why does stop replay result in a persistently high loss for these samples? Is the loss calculation also applied to the samples from the buffer? if the large loss represents forgetting, it should be the loss calculated only from buffer.
> > >
> > > 4. For the ABS scoring function, what if $z<0$ (more previous samples with negative scores)? And Equation (5) is still confusing
> > > $$
> > > p(x)  = \phi p_{curr}(x) + (1-\phi)p_{past}(x)
> > > $$
> > > For a sample from the past, say $x \in D_{<t}$, how do you compute $p(x)$?
> > >
> > > The experimental results are promising, but some details of the paper are still not clear, and the presentation can be improved. Given my above concerns, I will keep my score.

---

> ### Author Response · Authors · 2023-11-23
>
> Thank you kindly for your response.
>
> 1) We apologize for the confusion regarding our statement on underfitting. Indeed, our choice of benchmark already includes classification with more than 10 classes (i.e., CIFAR-100, NTU-60, and Food 101-N). Instead, we referred to "underfitting" as the phenomenon linked to the restricted amount of epochs allowed in the online noisy CL setting, which, in our opinion, limits the applicability of current methods in real-world scenarios.
>
> As for **Food-101N**, we kindly refer the reviewer to check carefully Sec. 4.1 (pag. 6) of the revised manuscript, as well as the results in Tab. 3 (pag. 7).
>
> 2) We avoided adopting PuriDivER as a starting point for our method as our intention is not to extend existing noisy online CL methods in the multi-epoch scenario, but rather to adapt current multi-epoch CL for a noisy scenario.
>
> For what concerns the consolidation stage, we posit that its omission constitutes a strength of our method. Specifically, current online CL methods require a consolidation stage to avoid completely underfitting the current task. However, their effectiveness remains limited as this stage relies on finetuning only on data from the buffer at the end of each CL task. Instead, our method surpasses almost all consolidation-based methods without the need for any consolidation, with a notable gain in training time (on CIFAR-10 with 40% noise, our method takes about 1h30m to complete, while PuriDivER and PuriDivER.ME take 5h15m and 2h50m respectively).
>
> For what concerns the weakness raised regarding the presentation of **PuriDivER's consolidation**, we agree with the reviewer and will include a brief description **in the revised manuscript**.
>
> 3) Yes, as stated in the introduction:
>    "We depict the loss trend for both the clean and noisy samples **in a memory buffer** produced by a rehearsal baseline ..."
> Following the reviewer's comment, we will update the caption to include such detail to avoid future confusion.
>
> 4) To answer the reviewer's first part of the question, since $z$ is the same for both sides of the dis-equation, the imbalance would still persist, although it would favour the opposite case with samples from the past being selected for removal disproportionally. We argue that this motivates our choice of designing ABS. As for the second part, we would like to point out that $p(\mathbf{x})$ defines the distribution from which to sample the example from the buffer to replace. To do this while avoiding the aforementioned problem of balancing the buffer between samples from the present and the past, we rely on a two-stage algorithm. Specifically, we initially first sample from the binomial distribution $\phi$ (with probability $q$ = $\frac{M_{curr}}{M}$ if we wish to select a sample from the present (i.e., $p_{curr}$) or the past (i.e., $p_{past}$). The latter scores are defined as:
> $p_{curr}(x) = \frac{L(x)}{z_{curr}}$, with $z_{curr}=\sum_{x\in M_{curr}}$
> $p_{past}(x) = \frac{L(x)}{z_{past}}$, with $z_{past}=\sum_{x\in M_{past}}$
> Thus, following the reviewer's question, if we choose to replace a sample from the past, we will sample $x\sim p_{past}$. In the original manuscript, for brevity, we summarized the above solution as sampling from $p(x) = \phi p_{curr}(x) + (1-\phi) p_{past}(x)$, with $\phi$ being used as a binary selector. However, we agree with the reviewer that such a form might be confusing, thus we will extend the explanation of ABS in the revised manuscript.
>
> We deeply value the reviewer's feedback on our paper. We've addressed every concern they raised, and earnestly request them to engage in a discussion with us. If they find our revisions satisfactory, we hope for a revised score reflecting the improvements, as such a gesture could encourage other inactive reviewers to engage in the discussion.

---

> > ### Comment · Reviewer_aFAe · 2023-11-23
> > **Response**
> >
> > Thanks for the authors' extensive response
> >
> > - For the Food-101N dataset, I am interested in the statistics of the dataset itself, e.g. How many classes? How many samples for each classes? Are these also from website images? Is the noise type of it similar as that of WebVision? I believe this is crucial to show the potentials of the research directions. Although there are classification tasks with more than 10 classes, but those are on synthesized benchmarks. I believe the this may also resolve other reviewer's concern that if noise label problem is a worthy question in CL and may show that the proposed algorithm can truly help in real world scenario.
> >
> > - For the scoring function, the extra explanation is extremely helpful! The original formulation $p(x) = \phi p_{curr}(x)+(1-\phi)p_{past}(x)$ is actually wrong unless you define $p_{curr}$ and $p_{past}$ for both past samples and current samples. But the revised on is clear and precise. Upon this, I am willing to increase my score.

---

> > > ### Author Response · Authors · 2023-11-23
> > >
> > > We deeply thank the reviewer for their encouraging feedback and their suggestion.
> > >
> > > Regarding the experiments on Food-101N, we follow the protocol of PuriDivER and split the 101 classes into 5 tasks (the first 4 containing 20 classes and the last containing 21). Each class is relatively balanced as it contains around 523 images (with a standard deviation of around 11). We train a ResNet34 both Food-101N and WebVision, and both datasets are estimated to contain a similar amount of instance-level label noise (20% according to [survey]), containing images crawled from the web.
> > >
> > > Following the reviewer's concern, we will include such details in the supplementary material.
> > >
> > > [survey]: Hwanjun Song, Minseok Kim, Dongmin Park, Yooju Shin, and Jae-Gil Lee. Learning from noisy labels with deep neural networks: A survey. IEEE Transactions on Neural Networks and Learning Systems, 2022

---

### Official Review · Reviewer_2XeP · 2023-11-01

**Soundness:** 2 fair
**Presentation:** 2 fair
**Contribution:** 2 fair
**Rating:** 5
**Confidence:** 3

**Summary:**

This paper focuses on continual learning with noisy labels and mitigating the forgetting of knowledge of past learning tasks. To achieve the two targets, this paper proposes to apply the experience replay method and proposes a new sample selection strategy called Asymmetric Balanced Sampling (ABS) for replay. ABS aims to select clean samples with lower loss from the current task and select the most informative samples with higher loss from the past tasks. This paper claims that the clean samples can increase the purity of buffer and the informative samples can increase the diversity to better mitigate forgetting. To evaluate the effectiveness of applying the replay with ABS on the tasks of continual learning with noisy labels, this paper set up experiments on 4 different datasets with different noisy ratios. The results prove the methods can achieve better performances on all classification tasks compared with other CL methods with different noisy label learning approaches.  Additional experiments also present the ABS can outperform other sample methods.

**Strengths:**

1. This paper focuses on the problem of continual learning with noisy labels, which is vital in real scenarios.
2. This paper applies experience replay into noisy label problems and proposes novel sampling methods for the replay.
3. The experiments are comprehensive, including four different types of datasets
and two different noise injection processes. This paper compares the
proposed methods with other different works of continual learning and noisy
label learning. The improvement of the proposed method is significant.

**Weaknesses:**

1. The presentation of the part of the method requires more details. For example, what does the task D_t = (X_t, Y_t) mean? Does it mean a dataset or distribution? And what does D_t ∩ M mean in eq.(4)?
2. The experiments only provide an average accuracy for all the tasks and do not prove the proposed method can reduce the forgetting of past tasks directly.

**Questions:**

a) How to guarantee the stored data in the buffer does not contain a noisy label when selecting the data with high losses.
b) How does the size of the buffer impact the performance?
c) How does the proposed method perform if the order of sequence is shuffled?
d) When the length of a sequence is increased, more informative data is added and how to preserve the information from the first several tasks with a smaller size of data in the buffer?

---

> ### Author Response · Authors · 2023-11-16
> **Response to the questions**
>
> We kindly thank the reviewer for their suggestions and the strengths highlighted. We encourage the reviewer to have a further discussion to help us improve the paper and the rating.
>
> **Question (a)**:  We ground our technique on the findings in both [SSF] and [toneva2018ans], which **demonstrate** that both noisy and complex samples exhibit a stronger tendency to be forgotten and can be distinguished based on their loss, as well as empirical findings of most of the works on Leaning with Noisy Labels (LNL). Indeed, in most cases, LNL methods successfully rely on the small-loss criterion as a technique for separating clean data [Dividemix, CoTeaching, PuriDivER]. Additionally, we conducted empirical evaluations to ensure the same behaviour occurs in a continual learning scenario, as demonstrated in the loss separation depicted in Fig. 1(a). While we cannot completely guarantee the absence of noise in the buffer, the effectiveness of our method in preserving non-noisy data within the buffer demonstrates its improved robustness, as well as subsequent evaluations on the cleanliness of buffer data shown in Fig. 6 of the main paper.
> **Question (b)**: Regarding the question about how the size of the buffer impacts the performances, we argue that in a scenario where it is crucial to maintain a clean buffer for accurate memory of past tasks, having a very large buffer, as is common in cases aimed at mitigating forgetting, might have a small impact on performance due to the poisoning effect of noise in the data stored. Instead, in such a scenario it is far more effective to extract a smaller amount of clean samples. To demonstrate this, we conduct a small evaluation of the hardest scenarios of the main table to investigate the effect of different buffer sizes. Such an evaluation shows that the second-best methods we evaluated (i.e., our ad-hoc version of PuriDivER, PuriDivER.ME), cannot effectively take advantage of the increased buffer, whereas our proposal can, and remains competitive across the board. The Final Average Accuracy results to assess the just-mentioned evaluations are reported in the following table.
> | Benchmark          | Seq. CIFAR-100 | Seq. CIFAR-100 | Seq. CIFAR-100 | Seq. CIFAR-100 | Seq. CIFAR-100 | Seq. CIFAR-100 | Seq. CIFAR-100 | Seq. CIFAR-100 | Seq. CIFAR-100 |
> |--------------------|-----|-----|-----|-----|-----|-----|-----|-----|-----|
> | Buffer size        | 500 | 500 | 500 | 2000 | 2000 | 2000 | 5000 | 5000 | 5000 |
> | | symm | symm | asymm | symm | symm | asymm | symm | symm | asymm |
> | Noise rate         | 40% | 60% | 40% | 40% | 60% | 40% | 40% | 60% | 40% |
> | PuriDivER.ME       | $23.72\pm0.48$ | $18.87\pm0.83$ | $20.40\pm0.98$ | $37.61\pm0.85$ | $27.18\pm0.76$ | $30.22\pm0.74$ | $39.48\pm1.2$ | $22.62\pm0.71$ | $32.07\pm1.18$ |
> | OURs               | $27.48\pm0.35$ | $17.50\pm0.55$ | $21.62\pm0.48$ | $38.64\pm0.57$ | $26.34\pm0.85$ | $29.26\pm0.91$ | $45.12\pm0.30$ | $33.80\pm0.37$ | $\mathbf{35.38\pm1.31}$ |
> | w. buffer fit.     | $\mathbf{30.32\pm0.97}$ | $\mathbf{19.24\pm0.74}$ | $\mathbf{22.66\pm0.80}$ | $\mathbf{41.58\pm0.63}$ | $\mathbf{30.13\pm0.98}$ | $\mathbf{31.49\pm0.38}$ | $\mathbf{47.63\pm0.66}$ | $\mathbf{35.46\pm0.53}$ | $34.61\pm1.04$ |
>
> **Question (c)**: Following the reviewer's suggestion, in the following table, we present the accuracy averaged over 3 runs with 3 different random shuffles. The results are on par with those originally reported, exhibiting a relatively modest standard deviation. This sustains the finding that, regardless of the method used, the choice of class order has a negligible impact on performances.
> | Benchmark   | Seq. CIFAR-100 | Seq. CIFAR-100 |
> |-------------|----------------|----------------|
> |             | symm           | asymm          |
> | Noise rate  | 60             | 40             |
> | OURs        | $\mathbf{28.60\pm0.55}$ | $\mathbf{31.32\pm1.35}$ |
> | PuriDivER.ME   | $15.56\pm0.46$ | $23.22\pm0.50$ |
>
> **Question (d)**:Most rehearsal methods rely on the $reservoir$ mechanism, which is designed to yield an i.i.d. representation of the original distribution. This, combined with our selection strategy, which guarantees the data is both **clean** and **class-balanced**, leads to strong performances of the method even as the number of tasks increases.

---

> ### Author Response · Authors · 2023-11-16
> **Comments on the weaknesses**
>
> **Weakness 1**: In our paper, $D_t$ was the set of data belonging to the specific $t^{th}$ task, containing $(X_t , Y_t)$ tuples of image and label. Consequently, in Eq. (4),  $D_t \cap M$ is the intersection between current task data and buffer data, so $ (x_t,y_t) \sim D_t \cap M$ stands for data belonging to both sets.
> We acknowledge that the $\textit{problem setting}$ is crucial for method comprehension, and we understand the importance of providing detailed information for those who have no knowledge about the continual learning setting. Following the reviewer's suggestion, we therefore rearranged Sec. 3.1 to enhance comprehensibility and now treat $D_t$ as a distribution. Specifically, the new version of the sentence is now  "We formalize the problem of **Continual Learning under Noisy labels** (**CLN**) as learning from a sequence of $T$ tasks. During each task $t \in \{0, 1, \dots, T\}$, input samples $\mathbf{X_t}$ and their corresponding annotations $\mathbf{Y_t}$ are drawn from an i.i.d. distribution $\mathcal{D}_t$."
>
> **Weakness 2**: We have opted for the most commonly used forgetting metric in the Continual Learning field (see Eq.(1) of the submitted supplementary material), applying this measure to each conducted experiment. As mentioned in Sec. 4.1 of the main paper, we kindly invite the readers to examine the **supplementary materials** provided alongside the main paper, as the evaluations they consider lacking are documented there. Moreover, as reported in Tab. D of the supplementary material, our method outperforms the others in terms of **Final Forgetting** metrics.
>
> [SSF]: Maini, Pratyush, et al. "Characterizing datapoints via second409 split forgetting." In aNeurIPS, 2022.
> [toneva2018an]: Toneva, Mariya, et al. "An empirical study of example forgetting during deep neural network learning." In ICLR Workshop, 2019.
> [DivideMix]: Li, Junnan, et al. "Dividemix: Learning with noisy labels as semi-supervised learning." In ICLR Workshop, 2020.          [CoTeaching]: Han, Bo, et al. "Co-teaching: Robust training of deep neural networks with extremely noisy labels." In aNeuIPS. 2018.
> [PuriDivER]: Bang, Jihwan, et al. "Online continual learning on a contaminated data stream with blurry task boundaries." In CVPR. 2022.

---

### Author Response · Authors · 2023-11-20
**Thanks to the reviewers; awaiting responses**

We kindly thank the reviewers for all their comments and feedback, thanks to which we had the chance to enhance our manuscript and strengthen our evaluations.

We are pleased that each reviewer, except for QKiJ, has valued the contribution of our method for its novelty, for being well-presented and motivated, for the significant improvements w.r.t. other methods and for the prominence of the setting.

Following the reviewers' comments, we provide a second real-world noisy dataset. Precisely, we evaluate our model on the Food101-N [F101] dataset, following the setting of PuriDivER. We will include these and/or other results, among those that the reviewers deem useful, in the main paper.

We strongly encourage reviewers to engage in discussions and take the time to provide additional explanations, to give us a chance to improve the paper scores and ratings.

[F101]: Lee, Kuang-Huei, et al. "Cleannet: Transfer learning for scalable image classifier training with label noise." Proceedings of the IEEE conference on computer vision and pattern recognition. 2018.

---

> ### Author Response · Authors · 2023-11-21
> **Manuscript updated**
>
> We have submitted a revised version of the manuscript incorporating improvements based on the suggestions provided by the reviewers. We look forward to the reviewers having the opportunity to check the updated manuscript along with our responses to their comments and questions.

---

### Author Response · Authors · 2023-11-23
**New version of the manuscript**

We thank all the reviewers involved in the rebuttal, as their suggestions allowed us to improve our manuscript. In this regard, we uploaded a newer version of both the manuscript and supplementary material with:
- further details about what's depicted in **Fig. 1**
- brief mention of PuriDivER's consolidation (**Sec. 4.4**)
- further details regarding the Asymmetric Balanced Sampling procedure (**Sec. 3.3**)

---

### Meta-Review · Area_Chair_bdsq · 2023-12-01

**Metareview:**

This paper worked on continual learning with noisy labels (a combination of continual learning and noisy-label learning) and proposed two methods called alternate experience replay and asymmetric balanced sampling to mitigate forgetting in continual learning and mitigate memorization of mislabeled data in noisy-label learning. The major issue is its limited novelty as pointed out by our reviewers. Moreover, some technical questions have not been well addressed, for instance,
> How to guarantee the stored data in the buffer does not contain a noisy label when selecting the data with high losses?

from Reviewer 2XeP. After the rebuttal, the only positive reviewer said that
> I only gain insights after iterative discussions with the authors, and my accept is based on the authors' promise that they would further modify the paper based on the discussions. As such, I would not strongly recommend an acceptance to this paper.

Therefore, we think we cannot accept the paper for publication at ICLR 2024 (at least the current version is not ready enough).

**Justification For Why Not Higher Score:**

The major issue is its limited novelty as pointed out by our reviewers. Moreover, some technical questions have not been well addressed.

**Justification For Why Not Lower Score:**

N/A

---

### Decision · Program_Chairs · 2024-01-16

Reject